# Mechanism Underlying Naringenin Hypocholesterolemic Effects: Involvement of Estrogen Receptor α Subtype

**DOI:** 10.3390/ijms232415809

**Published:** 2022-12-13

**Authors:** Valentina Pallottini, Marco Segatto, Filippo Acconcia, Marco Fiocchetti, Maria Marino

**Affiliations:** 1Department of Science, Section Biomedical Science and Technology, University Roma Tre, Viale Marconi 446, 00146 Rome, Italy; 2Neuroendocrinology Metabolism and Neuropharmacology Unit, IRCSS Fondazione Santa Lucia, Via del Fosso Fiorano 64, 00143 Rome, Italy; 3Department of Biosciences and Territory, University of Molise, Contrada Fonte Lappone, 86090 Pesche, Italy

**Keywords:** cholesterol, estrogen receptor α, HepG2 cells, hypocholesterolemic effect, naringenin

## Abstract

Naringenin (Nar) is one of major citrus flavonoids predominantly found in grapefruit and orange. In vivo studies have demonstrated Nar potential as a normolipidemic agent capable to reduce circulating cholesterol in hypercholesterolemic rabbits, rats, and patients, suggesting a new role for this molecule in cardiovascular disease prevention. Although Nar cholesterol-lowering effects are known, the underlying mechanisms have not yet been elucidated. Interestingly, Nar binds to the estrogen receptors (ERs), modulating both transcriptional and membrane-initiating signals. Although estrogen and ERs are deeply involved in lipid metabolism, no data are available regarding a putative role of these nuclear receptors as mediators of the hypocholesterolemic effect exerted by Nar. Thus, the aim of this work was to study the involvement of ERs in Nar-induced modulation of cholesterol metabolism. Results obtained in HepG2 cell line demonstrate that Nar can modulate the molecular network of cholesterol homeostasis. However, these effects were only partially dependent on the activity of estrogen receptor α. As a whole, our data highlight new molecular mechanisms by which Nar influences cholesterol metabolism, opening a new scenery about dietary impact on human health.

## 1. Introduction

Coronary heart disease (CHD) is the major cause of morbidity and mortality in Western industrialized countries. Every year, cardiovascular disease (CVD) kills approximately 16 million people. In the US, 43% of deaths are attributable to CVD, and in 2019, an estimated 558,000 deaths were caused by coronary heart disease, and 109,000 deaths were caused by ischemic stroke [1].

A positive correlation among elevated blood levels of low-density lipoprotein cholesterol (LDL-C), atherosclerosis, and coronary heart disease has been assessed; therefore, dyslipidemia is one of the main risk factors leading to CHD. Prevention of CHD remains a goal for researchers working with health or pharmaceutical care. In primary prevention, several modifiable risk factors should be considered for controlling this disease, such as the combination of smoking, hypertension, hyperlipidemia, obesity, sedentary life, and dyslipidemia. However, it is quite difficult to control cholesterol levels and meet the aggressive targets for plasma LDL-C concentrations by modifying daily habits alone. Thus, pharmacological intervention is often necessary. Three main classes of compounds are used in current clinical practice to treat dyslipidemia: statins, a primary therapeutic tool, fibrates, monoclonal antibodies, synthetic interfering RNA (siRNA), and bile-acid sequestrants or resins [2,3]. Because of the seriousness of the illness, the potentially huge patient population, and the several side effects of current medications [4,5], discovery and development of new cholesterol lowering agents with different sites or action mechanisms from those of existing drugs are demanding for researchers both in academia and industry. 

It has been recognized that a high intake of fruits and vegetables protects against chronic diseases. Among fruits and vegetables, the protective relationship between the consumption of citrus fruits or juices and the risk of ischemic stroke has been demonstrated [6]. In addition, it has been reported that high consumption of citrus juice or grapefruit improved blood lipid profile in hyperlipidemic humans [7]. Citrus juices, especially orange juice and grapefruit juice are rich source of flavonoids: naringenin (Nar) being the main component. In liver-derived human cell lines (i.e., HepG2 cells), Nar was shown to reduce the secretion of very low-density lipoprotein (VLDL) [8,9] through the inhibition of acyl-coenzyme A:cholesterol acylTransferase (ACAT2) [8] and microsomal triglyceride transfer protein (MTP) [10,11], enzymes critical for VLDL assembly. Nar was also shown to induce LDL receptor transcription through phosphatidyl inositol 3 kinase (PI3K) activation upstream of sterol regulatory element-binding protein 1a (SREBP-1a), a transcription factor involved in lipid metabolism [8,11]. Other studies demonstrated that Nar activated enzymes important in fatty acid oxidation such as CYP4A1, whereas the same compound inhibited hydroxy methyl-glutaryl coenzyme A reductase (HMGCR), the key and rate-limiting enzyme of cholesterol biosynthetic pathway [12]. Interestingly, the Nar hypocholesterolemic effects may be determined by the direct HMGCR inhibition. Indeed, in vitro it has been shown that at millimolar doses, Nar directly interacts with HMGCR catalytic site with affinities similar to those observed for statins [13]. However, the precise molecular mechanisms underlying Nar effects are not fully elucidated. In addition to the ability of high Nar concentrations to interact with enzymes, the flavanone biological effects have been ascribed to different molecular mechanisms, which include the binding to the estrogen receptors (ERs) [14]. Epidemiologic studies have shown that prior to menopause, women are at a lower risk of developing CVD compared to men. Women also have decreased CVD-associated morbidity compared to age-matched men and develop CVD 10 years later than men [15]. Experiments on animal models have demonstrated that these cardioprotective effects of pre-menopausal status are mainly conferred by the action of circulating estrogen on the α subtype of the ERs (ERα). Loss of ERα increased the occurrence of atherosclerotic lesions through elevated serum cholesterol levels and increased high-density lipoprotein particle size [16]. Estrogen/ERα complex decreases blood pressure, promotes vasodilation, and reduces the proliferation and migration of vascular smooth muscle cells (VSMCs) [17,18]. Our previous studies demonstrated that Nar, at concentrations compatible with that present in the blood after a meal enriched with flavonoids, impaired ERα rapid signaling by interfering with ERα-mediated activation of ERK and PI3K pathways without any effects on the receptor transcriptional activities [14,19,20]. Estrogen receptors (ERs) are deeply involved in lipid metabolism [20,21,22,23], but there is no evidence regarding a putative role of these nuclear receptors as mediators of the hypocholesterolemic effect exerted by Nar. Thus, this work aims at studying the involvement of ERα in Nar-induced modulation of protein network implicated in cholesterol homeostasis. For this purpose, HepG2 cells, which express only ERα subtype [24,25], have been chosen as a well-established experimental model to study lipid metabolism. Notably, this cells line retains several human liver functions, including lipid synthesis and secretion of albumin, apoB, and lipoproteins in response to various nutritional or physiological stimulations [26,27].

## 2. Results

Nar effects on the main proteins committed to cholesterol homeostasis were assessed by studying the level of HMGCR, the key and rate-limiting enzyme of cholesterol biosynthetic pathway, that catalyzes the first committed step in cholesterol biosynthesis. HMGCR phosphorylation, which represents the enzyme short-term inhibition [28], and the expression of low-density lipoprotein receptor (LDLR) were also evaluated in cells treated with Nar. Twenty-four hours after Nar treatment, the HMGCR expression showed a dose-dependent reduction, whereas its phosphorylation state increased (Figure 1A), suggesting a reduction of intracellular cholesterol synthesis. On the other hand, already at 10^−7^ M, Nar induces the increase of LDLR, which is suggestive of an enhanced LDL cholesterol uptake (Figure 1B). 

The increased HMGCR phosphorylation state mediated by Nar prompted us to evaluate the proteins involved in the enzyme short-term regulation, and we selected 10^−6^ M Nar concentration since at this dose we already obtained the maximum effect on HMGCR. Thus, the level of phosphorylation (i.e., activation) of AMP-activated kinase (AMPK), the major HMGCR kinase at least in the liver, and the level of protein phosphatase 2A (PP2A) catalytic subunit, which activates HMGCR by de-phosphorylation [29,30], have been assessed. Nar stimulation (10^−6^ M for 24 h) increases AMPK phosphorylation state (Figure 2A) and, concurrently, it significantly reduces the levels of PP2A catalytic sub-unit (Figure 2B). 

These results prompted us to evaluate if Nar could also affect the key proteins controlling HMGCR and LDLR long-term regulation. Long-term regulation acts through transcriptional, translational, and post-translational control [31]. SREBP cleavage activating protein (Scap), an escort protein for SREBPs embedded in the endoplasmic reticulum [32], binds to SREBPs and escorts them from the ER to the Golgi apparatus where SREBPs are proteolytically processed to yield N-terminal active fragments (nSREBPs), which enter the nucleus and induce the expression of their target genes [28], in sterol-deprived cells. When cholesterol builds up in endoplasmic reticulum membranes, the migration of the Scap/SREBP complex is blocked, the proteolytic processing of SREBPs is abolished, and the transcription of the target genes declines. Endoplasmic reticulum retention of Scap/SREBP is mediated by sterol-dependent binding of Scap/SREBP to the endoplasmic reticulum resident Insig (INSulin Induced Gene) proteins [31]. Moreover, intracellular accumulation of sterols triggers binding of HMGCR to Insigs, which, in turn, initiate the ubiquitination and the subsequent proteasomal degradation of the enzyme [33]. 

Figure 3A shows that Nar treatment suppressed the production of the transcriptionally active fragment of SREBP-1, whereas no changes were observed in SREBP-2 (Figure 3A) or in Scap (Figure 3B). Furthermore, Nar administration also increased the expression of both Insig-1 and -2 proteins (Figure 3C). Taken together, these data suggest that Nar is not only able to modulate cholesterol metabolism by affecting the SREBP-1-dependent transcriptional regulation of genes committed to cholesterol metabolism, but it is also able to reduce HMGCR levels through Insig-dependent enzyme degradation [33].

The reduction of SREBP-1 mediated by Nar, together with the increase of Insig proteins, strongly corroborates the HMGCR decrease. Conversely, the LDLR increase cannot be explained by the observed variations of these regulatory proteins. Therefore, we checked the level of the proprotein convertase subtilisin kexin 9 (PCSK9), a 72-kDa protease highly expressed in the liver under the control of SREBPs. Once secreted, PCSK9 binds LDLR, inducing the redistribution of the receptor from the cell surface to lysosomes; indeed, PCSK9 appears to change the trafficking of the LDLR, diverting the internalized LDLR to degradation in lysosomes and preventing them from being recycled to the cell surface [34]. Figure 4 shows that PCSK9 is lower upon Nar treatment with respect to the control, suggesting that this polyphenol downregulates LDLR degradation. 

Because Nar modifies the protein network of cholesterol metabolism maintenance, we investigated the putative ERα involvement in these changes. Thus, the ERα pure antagonist ICI 182,780 was administered to HepG2 cells 15 min before Nar treatment and, subsequently, HMGCR protein content and phosphorylation state, as well as LDLR expression, were analyzed. Figure 5A shows that ICI 182,780 prevents Nar-induced HMGCR changes both at protein and phosphorylation levels; conversely, the ERα inhibition cannot prevent LDLR increase induced by Nar (Figure 5B). 

ICI 182,780 is an inhibitor of both ER subtypes; to investigate the specific involvement of ERα, subtype HepG2 cells were treated with Endoxifen (a selective ERα inhibitor) 15 min before Nar stimulation. Results show that, like ICI 182,780 treatment, Endoxifen pretreatment prevents Nar-induced HMGCR modulation (Figure 6A), whereas it was incapable of preventing LDLR changes (Figure 6B), confirming that ERα is the main ER subtype expressed in this cell line [25]. Thus, the following experiments were performed in the presence of ICI 182,780.

Nar-induced AMPK activation was hindered by ICI 182,780 administration (Figure 7A). On the contrary, the ER antagonist did not exert any effect on PP2A reduction mediated by Nar (Figure 7B). 

Intriguingly, the Nar effect on nSREBP1 and Insig-1 was independent on ERα (Figure 8A,B), whereas Nar-induced Insig-2 modulation depends on ER activation (Figure 8C).

## 3. Discussion

Eating habits and metabolic disorders are associated with negative effects on the risk factors for cardiovascular diseases, as they can induce increased cholesterol and triglycerides plasma levels. In the last years, natural food components have attracted considerable interest in treating hypercholesterolemia. In general, plant foods provide support for cardiovascular disease prevention due to their low amount of fats and the presence of bioactive compounds, which may act as cholesterol-lowering agents through different mechanisms. Among these molecules, Nar shows hypolipidemic properties both in vivo and in vitro [8,35,36]. Its potential as a normolipidemic agent have been largely established [9,10,11,12,37,38], and a role of PPARα, PPARγ, and LXRα in Nar-dependent lipid lowering has been also demonstrated [39]. Moreover, docking analysis provided evidence that Nar can bind to the active site of HMGCR and compete with the ligand HMG-CoA, thus competitively inhibiting enzyme activity and reducing cholesterol intracellular synthesis [40]. More recently, another report highlights the direct inhibitory activity of Nar; notably, this compound anchors to HMGCR with modes and affinities like those observed for statins [13]. Despite this evidence, it has been postulated that the hypocholesterolemic activity of Nar may be exerted at multiple levels, but the exact molecular mechanisms are still unclear and deserve further investigation [36,41]. Here, we evaluated the Nar-induced modulations of the protein network controlling cholesterol homeostasis. Additionally, since Nar binds to the ERs [14,19,20], the putative implication of ERα in the hypocholesterolemic effects induced by Nar has also been studied. Thus, we chose as experimental model HepG2 cells, which express only ERα subtype [25]. 

Our results show that Nar administration to HepG2 cell line reduces HMGCR activation state and increases LDLR protein expression, justifying the in vivo lipid-lowering properties exerted by this flavanone [7,37,38,39]. Specifically, Nar affects HMGCR activation state by modulating AMPK and PP2A, the two main proteins implicated in its short-term regulation. Nar also influences HMGCR protein levels, which were decreased upon 24 h treatment. In this context, Nar could govern the long-term regulation of the enzyme not only by suppressing the SREBP-1-mediated transcription, but also probably by enhancing HMGCR degradation through Insig proteins, which were significantly upregulated. Since SREBP-1 isoform is also committed to the regulation of genes involved in fatty acids metabolism [42], the reduction of its transcriptionally active fragment observed in this work further supports the notion that Nar decreases triglyceride content, as already extensively demonstrated [36]. The fact that the level of SREBP-2, the transcription factor devoted to LDLR gene activation, is not modified by Nar administration sustains the hypothesis that the increased level of LDLR could be ascribable to a decreased level of PCSK9, a protein produced by the liver and deeply involved in LDLR degradation [43]. Our results demonstrate that Nar administration can reduce PCSK9 levels, opening a new scenery to the pharmacological intervention on hypercholesterolemia. Indeed, PCSK9 inhibition is considered an attractive target for therapy against hypercholesterolemia. To date, the therapeutic avenues tested to pharmacologically inhibit PCSK9 in humans are mainly focused on gene silencing that targets both PCSK9 intra- and extra-cellular functions and on mimetic peptides and monoclonal antibodies that exclusively target circulating PCSK9 and, therefore, its extracellular function [44]. Thus, Nar-induced PCSK9 reduction may represent a prospective approach to effectively reduce hypercholesterolemia, avoiding the side effects often complained by statin users [4,45] and the high costs of monoclonal antibodies.

To focus our attention on the putative role exerted by ERs in Nar-induced modulation of proteins belonging to cholesterol regulatory network, we used a pharmacological approach by using ICI 182,780, a pure ER antagonist, and Endoxifen, which specifically inhibits ERα. The results demonstrate that, differently from LDLR, Nar-induced HMGCR regulation is dependent on ERs. Interestingly, the regulation of proteins involved in HMGCR phosphorylation, AMPK and PP2A, is dependent and independent on ERs, respectively. Turning the attention on long-term regulation, our results demonstrate that only Nar-dependent Insig-2 increase is influenced by ERs. This feature is supported by previous data establishing a role for Insig-2 as a link between estrogen protective effects and cholesterol homeostasis. In particular, alignment sequence analysis showed that rat Insig-2 gene presents an estrogen response elements-like (ERE-like) within intron 2, suggesting that Nar binding to ERs is able to increase Insig-2 protein level as well as estrogen [22]. To date, the literature data about the useful effects of Nar in the prevention of lipid related pathologies support the concept that the protective activity of this flavanone is mostly due to its high antioxidant and anti-inflammatory effects [36] or through the action of PPARα, PPARγ, and LXRα [39]. However, these effects are obtained at high Nar concentrations ranging from 50 to 200 µM, a concentration difficult to obtain in the plasma after a meal enriched with this flavanone due to the extensive metabolic processes by gut and liver enzymes that transform polyphenol in more soluble compounds. The data collected in the present work, obtained at Nar concentration compatible with its presence in the blood, expand the knowledge about Nar lipid-lowering effects, putting in a new piece in the already available literature. This piece can be added to further effects on different dysfunctions related to metabolism, such as insulin sensitivity [46], artery hypertension [47], and, in general, in metabolic disorders such as obesity and diabetes [48].The hypocholesterolemic effects induced by the flavanone are strictly related to its capability in modulating the protein network of cholesterol homeostasis acting on HMGCR activity and, intriguingly, reducing PCSK9 levels, a pharmacological target to treat hypercholesterolemia that is attracting the efforts of many researchers working in the field. Moreover, we demonstrate for the first time that ERα is inserted in the machinery triggered by Nar to exert its hypocholesterolemic effects, ushering a new vision on diet impact to human health.

## 4. Materials and Methods

### 4.1. Materials

All materials used were obtained from commercial sources and of the highest quality available. All materials with no specified source are obtained from Merck (Milano, Italy).

### 4.2. Cell Culture

HepG2 cells were used as experimental model. Cells were routinely grown in air containing 5% CO_2_ in RPMI-1640 medium, containing 10% (*v*/*v*) fetal calf serum, L-glutamine (2 mM), gentamicin (0.1 mg/mL), and penicillin (100 U/mL). Cells were passaged every 3 days and media changed every 2 days. HepG2 cells were grown to 70% confluence in flasks then stimulated with vehicle (Control, DMSO/PBS, 1/10, *v*/*v*), Naringenin (10^−7^, 10^−6^, and 10^−5^ M), for 24 h in presence or in absence of pre-stimulation for 1 h with the ER inhibitor ICI, 182,780 (10^−6^ M) or the specific ERα inhibitor Endoxifen (10^−6^ M).

### 4.3. Western Blot Analysis

Cells were washed twice in phosphate buffer saline (PBS), harvested with trypsin (1%, *v*/*v*), and pelleted by centrifugation. Pellets were re-suspended in PBS containing 0.1 M sucrose, 0.05 M KCl, 0.04 M KH2PO4, 30 mM EDTA, pH 7.4, and sonicated. The protein concentration was determined by the Lowry method [49].

Samples, obtained as previously described, were solubilized in 0.25 mM Tris–HCl, (pH 6.8) containing 20% (*w*/*v*) SDS, protease inhibitor cocktail (Merck, Milano, Italy), then boiled for 2 min. Solubilized proteins (10 µg) were resolved by SDS–PAGE at 100 V for 1 h and then electrophoretically transferred to nitrocellulose for 80 min at 100 V and 48C. The nitrocellulose was treated with 3% (*w*/*v*) bovine serum albumin (BSA) in Tris Buffered Saline (20 mM Trizma base, 137 mM NaCl, 0.1%, *v*/*v*, Tween-20, pH 7.6) and then probed over night with the antibodies listed in Table 1. The nitrocellulose was stripped by Restore Western Blot Stripping Buffer (Pierce Chemical, Rockford, IL, USA) for 10 min at room temperature and then probed with anti-α tubulin. Antibody reaction was visualized with chemiolumionescence Western Blotting detection reagent (GE Healthcare, Little Chalfont, UK), as already reported [50].

### 4.4. Statistical Analysis

Each mean was derived from four different experiments performed in duplicate. The statistical analysis consisted of unpaired t test when two means were compared, or ANOVAs followed by the Tukey’s multiple comparisons test when more than two means were compared. *p* values under 0.05 were considered statistically significant. All the statistical analyses were performed using GraphPad Prism 7.00 (Graphpad Software Inc., La Jolla, CA, USA) software for Windows.

## Figures and Tables

**Figure 1 ijms-23-15809-f001:**
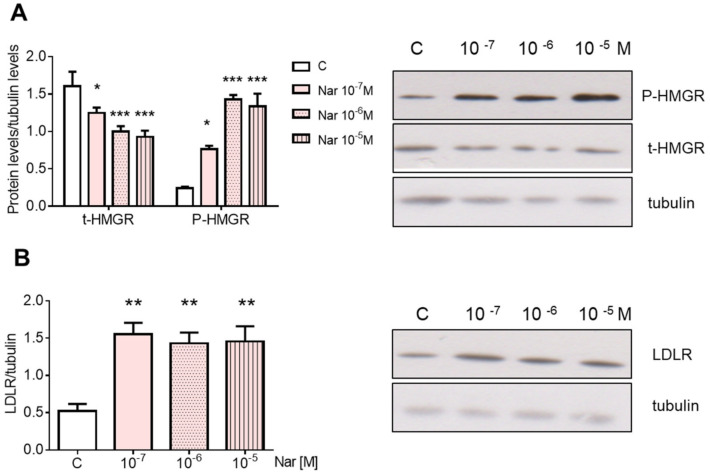
Naringenin dose-dependent effect on HMGCR and LDLR in HepG2 cells. Panel (**A**) shows the densitometric analyses (left) and a representative Western blot (right) of total and phosphorylated HMGCR after 24-h treatment with different doses of Nar. Panel (**B**) shows the densitometric analyses (left) and a representative Western blot (right) of LDLR after 24-h treatment with different doses of Nar. The control of protein loading was done with tubulin. The experiments were performed four times in duplicate. The statistical analysis was performed with ANOVA followed by the Tukey’s multiple comparisons test. * = *p* < 0.05 vs. C; ** = *p* < 0.01 vs. C; *** = *p* < 0.001 vs. C.

**Figure 2 ijms-23-15809-f002:**
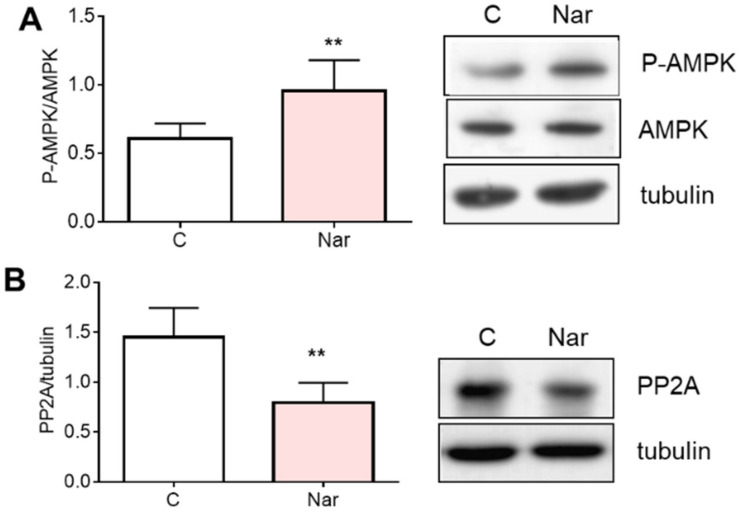
Naringenin effect on AMPK phosphorylation state and PP2A protein content in HepG2 cells. Cells have been treated for 24 h with 10^−6^ M Nar. Panel (**A**) represents, on the left, the densitometric analyses of AMPK phosphorylation state expressed as the ratio between P-AMPK and AMPK protein content and, on the right, a representative Western blot. Panel (**B**) shows the protein content of the catalytic subunit of PP2A. The control of protein loading was done with tubulin. The experiments were performed four times in duplicate. The statistical analysis was performed with unpaired Student’s *t* test. ** = *p* < 0.01 vs. C.

**Figure 3 ijms-23-15809-f003:**
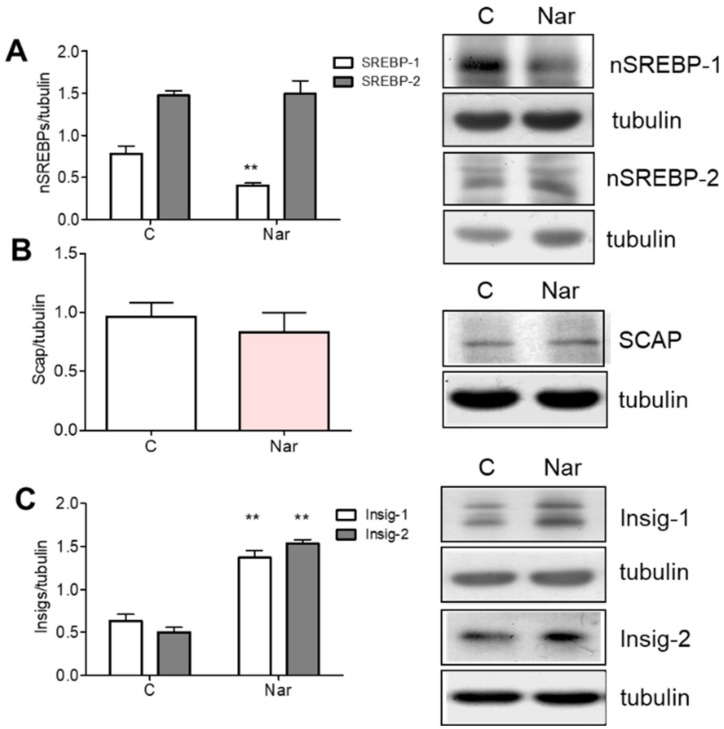
Naringenin effect on SREBPs, Scap, and Insigs protein content in HepG2 cells. HepG2 cells were treated for 24 h with 10^−6^ M Nar. The panels show, on the left, the densitometric analyses and, on the right, representative Western blots. Panels (**A**–**C**) illustrate the protein content of SREBP-1 and -2, Scap, and INsig-1 and -2, respectively. The control of protein loading was done with tubulin. The experiments were performed four times in duplicate. The statistical analysis was performed with unpaired Student’s *t* test. ** = *p* < 0.01 vs. C.

**Figure 4 ijms-23-15809-f004:**
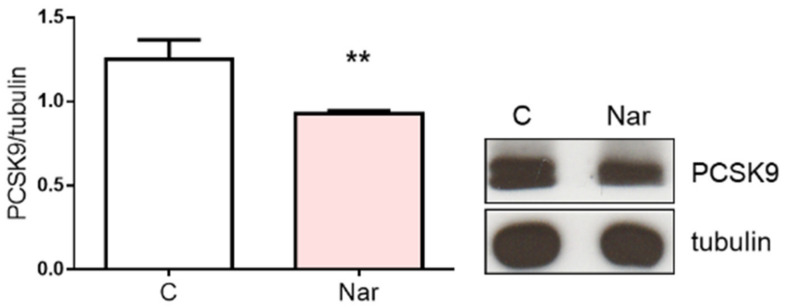
Naringenin effect on PCSK9 protein content in HepG2 cells. The figure shows the effects of 10^−6^ M Nar treatment for 24 h on PCSK9. On the left is illustrated the densitometric analysis, and on the right is a representative Western blot. The control of protein loading was done with tubulin. The experiments were performed four times in duplicate. The statistical analysis was performed with unpaired Student’s t test. ** = *p* < 0.01 vs. C.

**Figure 5 ijms-23-15809-f005:**
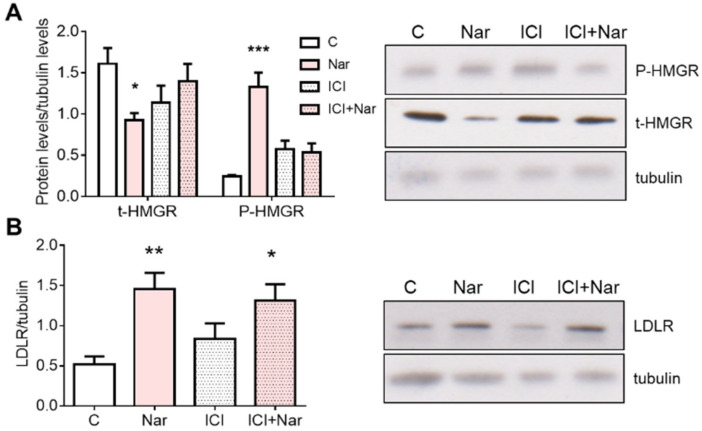
Involvement of ERs in Naringenin effects on HMGCR and LDLR in HepG2 cells. Panel (**A**) shows the densitometric analyses (left) and a representative Western blot (right) of total and phosphorylated HMGCR after 24-h treatment with 10^−6^ M Nar in presence and in absence of 10^−6^ M of the ER inhibitor ICI182,780 (ICI). Panel (**B**) shows the densitometric analyses (left) and a representative Western blot (right) of LDLR treated as already described. The control of protein loading was done with tubulin. The experiments were performed four times in duplicate. The statistical analysis was performed with ANOVAs followed by the Tukey’s multiple comparisons test. * = *p* < 0.05 vs. C; ** = *p* < 0.01 vs. C; *** = *p* < 0.001 vs. C.

**Figure 6 ijms-23-15809-f006:**
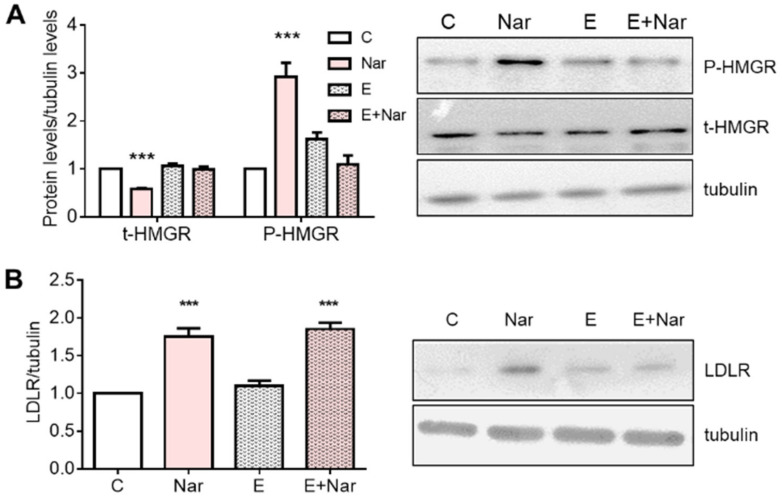
Involvement of ERα in Naringenin effects on HMGCR and LDLR in HepG2 cells. HepG2 cells were treated for 24 h with 10^−6^ M Nar in presence and in absence of the 10^−6^ M of the ERα inhibitor Endoxifen (E). Panel (**A**) shows the densitometric analyses (left) and a representative Western blot (right) of total and phosphorylated. Panel (**B**) shows the densitometric analyses (left) and a representative Western blot (right) of LDLR. The control of protein loading was done with tubulin. The experiments were performed four times in duplicate. The statistical analysis was performed with ANOVAs followed by the Tukey’s multiple comparisons test. *** = *p* < 0.001 vs. C.

**Figure 7 ijms-23-15809-f007:**
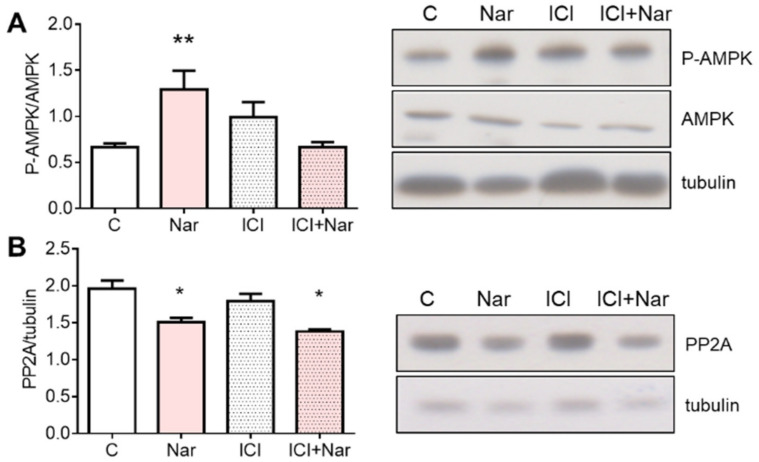
Involvement of ERα in Naringenin effects on AMPK phosphorylation state and PP2A in HepG2 cells. HepG2 cells were treated for 24 h with 10^−6^ M Nar in presence and in absence of the 10^−6^ M of the ER inhibitor ICI182,780 (ICI). Panel (**A**) represents, on the left, the densitometric analyses of AMPK phosphorylation state expressed as the ratio between P-AMPK and AMPK protein content and, on the right, a representative Western blot. Panel (**B**) shows the protein content of the catalytic subunit of PP2A. The control of protein loading was done with tubulin. The experiments were performed four times in duplicate. The statistical analysis was performed with ANOVAs followed by the Tukey’s multiple comparisons test. * = *p* < 0.05 vs. C; ** = *p* < 0.01 vs. C.

**Figure 8 ijms-23-15809-f008:**
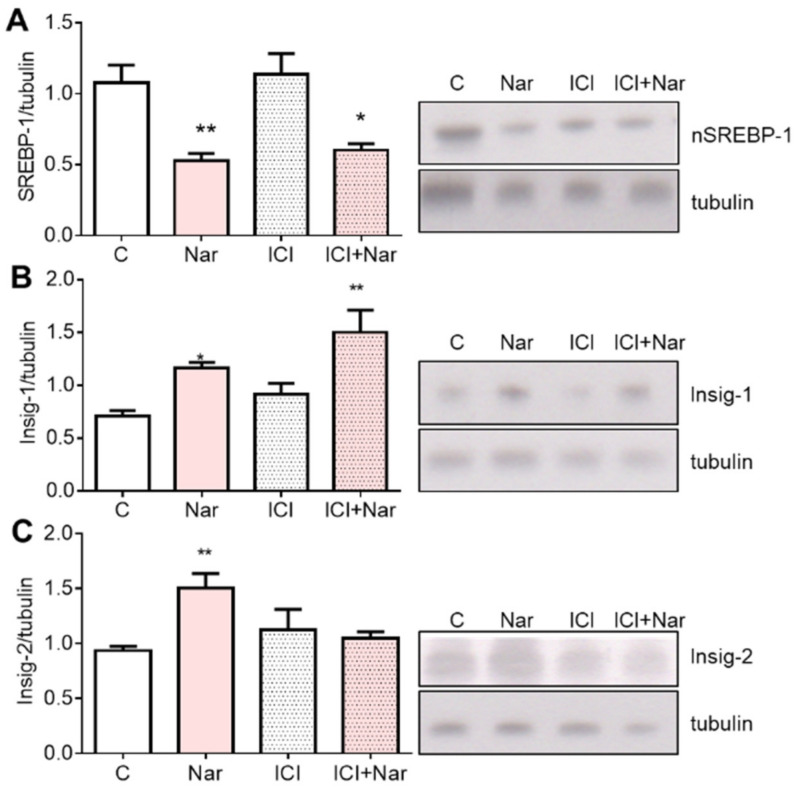
Involvement of ERs in Naringenin effects on SREBP-1, Insig-1, and -2 in HepG2 cells. HepG2 cells were treated for 24 h with 10^−6^ M Nar in presence and in absence of the 10^−6^ M of the ER inhibitor ICI182,780 (ICI). The panels show, on the left, the densitometric analyses and, on the right, representative Western blots. Panels (**A**–**C**) illustrate the protein content of SREBP-1, Insig-1, and -2, respectively. The control of protein loading was done with tubulin. The experiments were performed four times in duplicate. The statistical analysis was performed with ANOVAs followed by the Tukey’s multiple comparisons test. * = *p* < 0.05 vs. C; ** = *p* < 0.01 vs. C.

**Table 1 ijms-23-15809-t001:** List of antibodies used for western blotting assay.

Primary Antibody	Sources	Secondary Antibody	Sources
Anti-HMGCR	Upstate, Lake Placid, NY, USA	Goat anti-rabbit	UCS Diagnostic, Rome, Italy
Anti-Insig 1	Novus Biologicals, Littleton, CO, USA	Goat anti-rabbit	UCS Diagnostic, Rome, Italy
Anti-Insig 2	Santa Cruz Biotechnology, Santa Cruz, CA, USA	Rabbit anti-goat	Chemicon International, Temecula, Canada
Anti-SREBP 1 (N-terminal)	Abcam, Cambridge, UK	Goat anti-rabbit	UCS Diagnostic, Rome, Italy
Anti-SREBP 2 (N-terminal)	Abcam, Cambridge, UK	Goat anti-rabbit	UCS Diagnostic, Rome, Italy
Anti LDLR	Abcam, Cambridge, UK	Goat anti-rabbit	UCS Diagnostic, Rome, Italy
Anti PCSK9 (Anti NARC-1)	Santa Cruz Biotechnology, Santa Cruz, CA, USA	Goat anti-rabbit	UCS Diagnostic, Rome, Italy
Anti-tubulin	MP Biomedicals, Solon, OH, USA	Goat anti-mouse	UCS Diagnostic, Rome, Italy
Anti-AMPKα	Cell Signaling technology, Boston, MA USA	Goat anti-rabbit	UCS Diagnostic, Rome, Italy
Anti-P-AMPKα	Cell Signaling technology, Boston, MA USA	Goat anti-rabbit	UCS Diagnostic, Rome, Italy
Anti-Scap	Santa Cruz Biotechnology, Santa Cruz, CA, USA	Rabbit anti-goat	UCS Diagnostic, Rome, Italy
Anti-PP2A	Santa Cruz Biotechnology, Santa Cruz, CA, USA	Goat anti-rabbit	UCS Diagnostic, Rome, Italy

## Data Availability

The data presented in this study are available on request from the corresponding author.

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
