# Peer review of "Mechanism Underlying Naringenin Hypocholesterolemic Effects: Involvement of Estrogen Receptor α Subtype"

_ijms, 2022, doi:10.3390/ijms232415809_

Round 1
Reviewer 1 Report (Previous Reviewer 1)
The main goal of this paper is to investigate the mechanism of action related to the ability of naringenin, one of the most abundant flavonoids isolated from grapefruit and citrus juices, to modulate cholesterol homeostasis. Accordingly, the treatment with naringenin resulted in a decrease in HMGCR activation and an increase in LDLR protein expression, indicating that this could be the plausible mechanism of action implicated in the lipid-lowering properties of this flavonoid. This effect was also demonstrated to be partially dependent on the activity of the estrogen receptor α. The overall results appear promising, the paper falls within the journal's scope and is properly organized as a communication.
Only English minor spell check is required and some more references can be added (for example:
• J Med Food. (2020); DOI: 10.1089/jmf.2019.0216.
• Journal of Functional Foods (2019); DOI:10.1016/J.JFF.2018.11.030.
• Kidney Blood Press Res. (2022); DOI: 10.1159/000524172).
Author Response
Dear Editor,
We are pleased resubmit the revised version (R1) of the manuscript ijms-2080013. Minor revisions were required.
We would like to thank the reviewers for their comments which certainly improve our paper. We performed all the revision requested which are underlined in yellow in the manuscript. Following you can find a step-by-step detailed responses to reviewers’ comments.
Best regards Rome 8th December 2022
Valentina Pallottini
__________________________________________________________________
Reviewer #1
The main goal of this paper is to investigate the mechanism of action related to the ability of naringenin, one of the most abundant flavonoids isolated from grapefruit and citrus juices, to modulate cholesterol homeostasis. Accordingly, the treatment with naringenin resulted in a decrease in HMGCR activation and an increase in LDLR protein expression, indicating that this could be the plausible mechanism of action implicated in the lipid-lowering properties of this flavonoid. This effect was also demonstrated to be partially dependent on the activity of the estrogen receptor α. The overall results appear promising, the paper falls within the journal's scope and is properly organized as a communication.
Only English minor spell check is required and some more references can be added (for example:
- J Med Food. (2020); DOI: 10.1089/jmf.2019.0216.
- Journal of Functional Foods (2019); DOI:10.1016/J.JFF.2018.11.030.
- Kidney Blood Press Res. (2022); DOI: 10.1159/000524172).
Authors’ response: We thank the reviewer for the suggestions, we added the indicated references, moreover the paper has been read by a mother tongue colleague who amended some sentences.
Reviewer 2 Report (New Reviewer)
The manuscript ijms-2080013, which seems to be a revised submission, investigate the effect of low concentrations of naringenin on the expression of several proteins implicated in cholesterol homeostasis and how these effects could be mediated by naringenin binding to oestrogen receptor-alpha (Era). Overall, this is a meaningful work bringing new insights into the mechanism of hypocholesterolemic activity of naringenin.
The study was very well designed, experiments well performed and analysed, and conclusions are supported by the results. Overall, the authors reached to the conclusion that narigenin effect on ERa do not fully justify the modulation of cholesterol homeostasis by naringenin. Another interesting conclusion is that naringenin reduces PCSK9 levels. I wonder why this discovery is not highlighted in the abstract as it is in the conclusions.
Other minor revisions:
Figure1A (Western blot picture) – correct the concentration of naringenin 10-5 (-5 should be in superscript).
Figure 6 title. Correct the letter after ER.
Author Response
Dear Editor,
We are pleased resubmit the revised version (R1) of the manuscript ijms-2080013. Minor revisions were required.
We would like to thank the reviewers for their comments which certainly improve our paper. We performed all the revision requested which are underlined in yellow in the manuscript. Following you can find a step-by-step detailed responses to reviewers’ comments.
Best regards Rome 8th December 2022
Valentina Pallottini
______________________________________________________________________
Authors’ response: We thank the reviewer for the suggestions, we added the indicated references, moreover the paper has been read by a mother tongue colleague who amended some sentences.
Reviewer #2
The manuscript ijms-2080013, which seems to be a revised submission, investigate the effect of low concentrations of naringenin on the expression of several proteins implicated in cholesterol homeostasis and how these effects could be mediated by naringenin binding to oestrogen receptor-alpha (Era). Overall, this is a meaningful work bringing new insights into the mechanism of hypocholesterolemic activity of naringenin.
The study was very well designed, experiments well performed and analysed, and conclusions are supported by the results. Overall, the authors reached to the conclusion that narigenin effect on ERa do not fully justify the modulation of cholesterol homeostasis by naringenin. Another interesting conclusion is that naringenin reduces PCSK9 levels. I wonder why this discovery is not highlighted in the abstract as it is in the conclusions.
Other minor revisions:
Figure1A (Western blot picture) – correct the concentration of naringenin 10-5 (-5 should be in superscript).
Figure 6 title. Correct the letter after ER.
Authors’ response: We thank the reviewer for the suggestions and amended the indicated mistakes. Moreover, we highlighted in the abstract the PCSK9 reduction induced by Nar.
This manuscript is a resubmission of an earlier submission. The following is a list of the peer review reports and author responses from that submission.
Round 1
Reviewer 1 Report
The aim of this work is to demonstrate the capability of naringenin, one of the most abundant flavonoids isolated in citrus fruits and grapefruit, to modulate cholesterol homeostasis, through the ER α receptor.
The manuscript is well-written, the idea seems interesting, and the results obtained appear to be promising. However, the data are primarily the results of western blotting essays. Thus, in my opinion, the manuscript, in its present form, is not adequate to be published on IJMS. I recommend submitting the manuscript to a different and more appropriately selected journal or implementing the experimental section (computational studies, for example, could strengthen the results obtained).
Please note that the “Author Contributions” and “Data Availability Statement” sections have not been completed.
Author Response
Reviewer #1
The aim of this work is to demonstrate the capability of naringenin, one of the most abundant flavonoids isolated in citrus fruits and grapefruit, to modulate cholesterol homeostasis, through the ER α receptor.
The manuscript is well-written, the idea seems interesting, and the results obtained appear to be promising. However, the data are primarily the results of western blotting essays. Thus, in my opinion, the manuscript, in its present form, is not adequate to be published on IJMS. I recommend submitting the manuscript to a different and more appropriately selected journal or implementing the experimental section (computational studies, for example, could strengthen the results obtained).
Author response: We thank the reviewer for this comment. The paper has been submitted to a Special Issue entitled “Emerging Role of Lipids in Metabolism and Disease-Third Edition” that fits perfectly with the topic of our paper.
Please note that the “Author Contributions” and “Data Availability Statement” sections have not been completed.
Author response: We apologize for the mistake and completed the paraphs.
Reviewer 2 Report
The authors have investigated the mechanisms through which the citrus flavonoid, Naringenin (Nar), may influence cholesterol homeostasis. Using well established methodology (HepG2 cells and Western blotting) they have examined Nar treatment in combination with estrogen receptor (ER) blockade on a number of proteins involved in cholesterol biosynthesis to investigate the mechanisms that may contribute to the lipid lowering potential of citrus fruits/juices. The paper is well written and concisely arranged and their results have elegantly demonstrated the mechanisms involved in Nar action at physiologically achievable concentrations through its actions on the ERα.
The authors have undertaken simple but appropriate statistical analyses for the studies completed, within the paper all references to statistical significance are compared with the control group. I would be interested in adding the detail of any potential further interactions between Nar treatment and the combination of Nar and ICI or E inhibitors, and also any interactions: eg in Figure 5, Nar treatment significantly increase P-HMGR compared to control, however the graphed results also suggest that Nar treatment alone maybe different to that of E+Nar. If this interaction has been investigated but is not significant could reference to testing of this interaction be included in section 4.4-statisitical Analysis (Materials and methods). These interactions should also be investigated for the subsequent data where appropriate.
Minor grammatical errors for correction:
Lines 56/57: Citrus juices, especially orange juice and grapefruit juice are rich source of flavonoids being naringenin (Nar) the main component.
Suggested edit: Citrus juices, especially orange juice and grapefruit juice are rich source of flavonoids: naringenin (Nar) being the main component.
Lines 277-279: Our results demonstrate that Nar administration is capable to reduce PCSK9 level opening a new scenery to the pharmacological intervention on hypercholesterolemia.
Suggested edit: Our results demonstrate that Nar administration is capable of reducing PCSK9 levels, opening a new scenery to the pharmacological intervention on hypercholesterolemia.
Author Response
Reviewer #2
The authors have investigated the mechanisms through which the citrus flavonoid, Naringenin (Nar), may influence cholesterol homeostasis. Using well established methodology (HepG2 cells and Western blotting) they have examined Nar treatment in combination with estrogen receptor (ER) blockade on a number of proteins involved in cholesterol biosynthesis to investigate the mechanisms that may contribute to the lipid lowering potential of citrus fruits/juices. The paper is well written and concisely arranged and their results have elegantly demonstrated the mechanisms involved in Nar action at physiologically achievable concentrations through its actions on the ERα.
The authors have undertaken simple but appropriate statistical analyses for the studies completed, within the paper all references to statistical significance are compared with the control group. I would be interested in adding the detail of any potential further interactions between Nar treatment and the combination of Nar and ICI or E inhibitors, and also any interactions: eg in Figure 5, Nar treatment significantly increase P-HMGR compared to control, however the graphed results also suggest that Nar treatment alone maybe different to that of E+Nar. If this interaction has been investigated but is not significant could reference to testing of this interaction be included in section 4.4-statisitical Analysis (Materials and methods). These interactions should also be investigated for the subsequent data where appropriate.
Author response: We thank the reviewer for the insightful comment. We have added in Statistical analysis that we have performed a Tukey multiple comparison test, so we did the other comparisons among all the experimental groups, and what was statistically different was already indicated.
Minor grammatical errors for correction:
Lines 56/57: Citrus juices, especially orange juice and grapefruit juice are rich source of flavonoids being naringenin (Nar) the main component.
Suggested edit: Citrus juices, especially orange juice and grapefruit juice are rich source of flavonoids: naringenin (Nar) being the main component.
Lines 277-279: Our results demonstrate that Nar administration is capable to reduce PCSK9 level opening a new scenery to the pharmacological intervention on hypercholesterolemia.
Suggested edit: Our results demonstrate that Nar administration is capable of reducing PCSK9 levels, opening a new scenery to the pharmacological intervention on hypercholesterolemia.
Author response: We thank the reviewer for the suggestions. The manuscript has been amended accordingly.
Round 2
Reviewer 1 Report
Despite the manuscript fits with the special issue topic, I believe that the experimental part needs to be expanded if the manuscript is considered as an article to be published in IJMS. I suggest submitting it as a short communication.